# Production and Assessment of Poly(Lactic Acid) Matrix Composites Reinforced with Regenerated Cellulose Fibres for Fused Deposition Modelling

**DOI:** 10.3390/polym14193991

**Published:** 2022-09-23

**Authors:** Christian Gauss, Kim L. Pickering, Joshua Tshuma, John McDonald-Wharry

**Affiliations:** Division of Health, Engineering, Computing & Science, School of Engineering, The University of Waikato, Private Bag 3105, Hamilton 3216, New Zealand

**Keywords:** 3D printing technology, cellulose, mechanical properties, lyocell, bio-composites, PLA

## Abstract

Additive manufacturing can be a valuable tool to process polymeric composites reinforced with bio-based fibres, extending their use and opening new opportunities for more environmentally friendly materials. In this work, poly(lactic acid) (PLA) composites reinforced with regenerated cellulose fibres (lyocell) were processed into novel filaments and used for 3D printing. The Young’s modulus of the filaments increased with the addition of fibres, but substantial porosity was observed in formulations with 20 and 30 wt% of fibre content. Nonetheless, the composites were easily printed, and the formulation with 10 wt% of fibres presented the best tensile properties of 3D printed samples with average tensile strength, Young’s modulus, and strain at break of 64.2 MPa, 4.56 GPa, and 4.93%, respectively. It has been shown in this study that the printing process contributes to fibre alignment with small variations depending on the printing speed. Printed composite samples also had superior thermo-mechanical stability with a storage modulus up to 72 times higher than for neat PLA at 80 °C after the composite samples were heat-treated. In general, this work supports the potential use of regenerated cellulose fibres to reinforce PLA for 3D printing applications.

## 1. Introduction

Additive manufacturing (AM), or three-dimensional (3D) printing, is a collection of different technologies for the fabrication of 3D objects and can be used to produce complex geometries that would otherwise be difficult or impossible to produce using traditional manufacturing methods [1,2]. AM technologies are grouped into seven categories, depending on the type of material and printing method used: binder jetting, directed energy deposition, material extrusion, material jetting, powder bed fusion, sheet lamination, and vat photopolymerisation [3]. One of the most common methods is fused deposition modelling (FDM), also known as fused filament fabrication (FFF), which is based on material extrusion [4]. This type of printing technology has decreased in cost over the years, making it affordable enough for household use and increasing its popularity [5]. In the FDM process, 3D CAD models are sliced into layers and a generated G-code (Geometric code for computer numerical control) is transferred to the 3D printer that is fed with 1.75 or 3 mm thermoplastic filaments to print the object layer by layer.

Poly(lactic acid) (PLA) is one of the most commonly used bio-based thermoplastic polymers for 3D printing. It is already well consolidated in the market for its balanced combination of printability, mechanical properties, and cost [6]. PLA can be recycled and is a biodegradable polymer and is therefore ideal for components that need to be integrated into a circular economy [7,8]. However, the use of PLA in some applications is still limited due to its low thermo-mechanical stability and brittleness [9,10].

Fibre reinforcement has long been a method used for increasing the mechanical properties of thermoplastics used in 3D printing. Common materials used for fibre reinforcement are carbon and glass fibres [11,12,13]. However, these fibres limit the recyclability of the final filament and also require intensive resources during production. The use of cellulose-based fibres such as hemp, bamboo, flax, harakeke, wood fibres, and nano cellulose has been explored as a method of reinforcing PLA filaments without compromising their biodegradability, as well as reducing the environmental impact of fibre production [14,15,16,17,18,19,20,21,22,23,24]. Different production methods and compositions have been explored to produce these PLA-reinforced filaments. Optimum fibre contents between 1–30 wt% are reported depending on the type of fibre, resulting in 3D printed samples with ultimate tensile strength and Young’s moduli between 28–80 MPa and 2–7 GPa, respectively. Although in some cases good mechanical properties are achieved, processability issues often cause an increase in porosity and inadequate distribution of the fibres, which can decrease the mechanical properties in comparison with neat PLA. A detailed summary of the composition, production method, and tensile properties of PLA composites reinforced with cellulose-based fibres for FDM produced in the last 4 years is given in Appendix A. In addition, along with the reinforcement type that is used for PLA, the parameters that are used during the printing process have been found to have a significant influence on the mechanical properties of the final part [25,26,27,28,29,30]. Some of the printing parameters that are often considered in the literature are the nozzle temperature, print bed temperature, nozzle diameter, infill density, print orientation, layer height, feed rate and raster angle. When fibre-reinforced composites are used, especially with high fibre content (>20 wt%), non-uniform printing, nozzle blockage (clogging), and poor interlayer adhesion are generally reported, mainly due to the highly viscous nature of the composites [23,31,32]. Therefore, modification of the printing parameters is often necessary to achieve acceptable printing quality. Lyocell fibres, a type of regenerated cellulose, have been used to improve the overall mechanical properties of PLA composites produced by different methods, increasing tensile and flexural strengths, modulus of elasticity, strain at break, and impact strength [33,34,35,36]. These results show the advantage of these human-made bio-fibres over natural lignocellulose fibres, such as kenaf and hemp, the addition of which improves strength and Young’s moduli but may reduce toughness [34,35,37]. Lyocell fibres are produced using *N*-methyl morpholine-*N*-oxide (NMMO) in a process named lyocell to dissolve and regenerate cellulose into continuous fibres. There are five general steps in the lyocell process; dissolution, filtration, spinning, washing and finishing. As the fibres are bioderived and are produced in a closed system, they are often referred to as a sustainable fibre choice [38,39].

Existing research on PLA reinforced with lyocell fibres focuses mainly on injection or compression moulding processes [36,40,41], and to the best of our knowledge, there are no publications reporting the use of lyocell fibres for FDM. The combination of good mechanical properties, uniform dimensions of the fibres, and renewability make lyocell fibres an attractive choice for 3D printing in comparison to other bio-based fibres. Similar advantages observed in lyocell/PLA composites produced by conventional methods are expected in 3D printed objects. In this work we explored the use of commercial lyocell fibres to produce reinforced PLA composites for FDM, thoroughly characterising the processability and properties of the obtained filaments. The printability of these new composites, and resulting tensile properties, thermo-mechanical behaviour, and fibre alignment were also investigated and discussed.

## 2. Materials and Methods

### 2.1. Materials

Poly(lactic acid) (PLA) grade 2003D with melt flow index (MFI) of 6 g/10 min (210 °C, 2.16 kg) and specific gravity of 1.24 g·cm^−3^ was purchase from NatureWorks^®^, Plymouth, MN, USA. Lyocell fibres with a nominal length of 3 mm were provided by Lenzing^®^, Lenzing, Austria. The fibres had a nominal linear density of 1.7 dtex (equivalent to diameters between 10–12 µm) and a tenacity at break of 35 cN/tex (equivalent to approximately 536 MPa). Analytical grade Merck Millipore dichloromethane (DCM) was used for fibre extraction.

### 2.2. Characterisation of the Fibres

The thermal stability of the lyocell fibres and neat PLA used in this study were analysed through thermogravimetric analysis (TG) in a Perkin Elmer STA6000 TGA from 30 to 600 °C at a heating rate of 10 °C/min under argon flow of 40 mL/min. The fibres were also analysed in a Panalytical Empyrean XRD (Worcestershire, UK) using CuKα radiation (40 kV; 40 mA) equipped with a PixCel linear detector. The fibres were scanned in a 2θ range of 5°–45° using a scanning step of 0.01° and an equivalent exposure time of 40 s. The obtained X-ray diffraction pattern was analysed in the software HighScore^®^ Plus, Malvern Panalytical) (Worcestershire, UK) and submitted to a Rietveld refinement using the crystal structure of cellulose II [42]. The degree of crystallinity of cellulose was determined by including the amorphous phase during the refinement, as proposed by Nam et al. (2016) [43]. Crystallite sizes of planes (110) and (020) were calculated using the Scherrer equation [44].

### 2.3. Filament Production

Filaments with a nominal diameter of 1.75 mm of compositions presented in Table 1 were produced by extrusion. Before extrusion, the fibres were oven-dried at 103 °C for 24 h and then were vacuum dried with the PLA granules at 60 °C for 4 h. First, PLA and fibres were mixed in the correct proportion and manually fed into a Labtech LTE 20–44 twin-screw extruder, Praksa, Thailand, with an L/D ratio of 44 and processed at a temperature profile of 130, 180, 190, 190, 190, 190, 195, 195, 210 °C (from feed zone to extrusion die) using an exit die of 10 mm at a screw speed of 30 rpm. The obtained material was granulated into particles of approximately 3 mm in a Moretto GR granulator, Mercer County, PA, USA, and used to produce filaments in the same twin-screw extruder at the same temperature profile. During processing, a negative pressure of −100 to −200 mmHg was set at the end of the barrel using a vacuum pump. Filaments with a diameter of 1.75 ± 0.2 mm were produced using a 1.9 mm die at screw speeds between 18–36 rpm and feeding speeds between 4.4–7.4 rpm. The processing conditions for each formulation are given in Table 1. After defining a spooling speed (synchronised with the extrusion speed), the exit die pressure was maintained constant by adjusting the extrusion speed. The filaments were cooled down in air using a fan directed to the spooling system. A schematic representation of the setup used to produce the filaments is shown in Figure 1.

### 2.4. Characterisation of Filaments

#### 2.4.1. Microscopy

The filaments were observed using optical microscopy to check the fibre distribution and the presence of porosity. The samples were mounted in epoxy resin and submitted to rough and fine grinding, followed by polishing with Streuers OP-U silica suspension. The samples were then analysed in an Olympus model BX53 (Tokyo, Japan) optical microscope. The obtained micrographs were used to estimate the porosity percentage of the filaments using the ImageJ analysis software [45]. Ten micrographs were taken from two different filament samples of each formulation for the porosity quantification. Lower magnification images of the filaments were also taken using an Olympus stereo microscope (model SZX7, Tokyo, Japan).

#### 2.4.2. Tensile Testing

Tensile testing of the filaments was conducted in an Instron^®^, model 5982 (Norwood, MA, USA) universal mechanical testing machine with grips custom-made for 1.75 mm filaments. Six specimens per condition were tested using a 5 kN load cell at 5 mm/min. A 25 mm extensometer was used to calculate Young’s modulus. The extensometer was removed from the samples at a strain of 1% to avoid any influence on the filament tensile strength and strain at break. A constant length of 50 mm was adopted for the free span of the filaments, and the diameter of each sample was measured in three different positions using a digital calliper. Before testing, the filaments were conditioned in a conditioning chamber for at least 48 h at 23 °C and 50% relative humidity.

#### 2.4.3. Fibre Extraction and Analysis

In order to evaluate the effect of filament processing on the fibre morphology, the fibres were extracted from the composite by dissolving PLA in dichloromethane (DCM) followed by filtration. Samples of filaments and granules after the first extrusion were dissolved at room temperature in DCM under continuous stirring at a concentration of 10 g/L for 6 h. The obtained fibres were rinsed thrice with DCM and dried at room temperature for 48 h. The as-received and extracted fibres were then placed in a glycerol droplet between two microscopy glasses and analysed in an Olympus (model BX53) optical microscope. This process was conducted for the first extrusion process and the final filaments. The obtained micrographs were used to determine the length of the fibres using the ImageJ analysis software (Bethesda, MD, USA).

### 2.5. 3D Printing

Samples for tensile testing, dynamic mechanical analysis (DMA), and X-ray diffraction (XRD) analysis were 3D printed in a MakerGear™ M2 desktop 3D printer (Beachwood, OH, USA) using the Simplify 3D^®^ software package (Cincinnati, OH, USA) for slicing the CAD files and controlling the 3D printer. Before printing, all the filaments were vacuum dried at 50 °C. The samples were printed using the printing parameters given in Table 2.

The samples for tensile testing were printed using ASTM D638 [46] type V samples, with free-span nominal dimensions of 3.18, 3.20, and 10.96 mm for width, thickness, and length, respectively. The ASTM D638 type V sample design was modified by increasing the length of the specimen by 15% (printing direction) to adequately accommodate a 10 mm extensometer. Small disks with a diameter of 30 mm and thickness of 1.5 mm were 3D printed for X-ray diffraction (XRD) analysis using the same conditions as the samples for tensile testing but using different printing speeds (1800 and 4800 mm/min). Samples with 5 × 1.5 × 30 mm^3^ were printed for dynamic mechanical analysis (DMA). The dimensions of all the samples and corresponding printing directions are represented in Appendix A. After 3D printing, all the samples were conditioned in a conditioning chamber for 48 h at 23 °C and relative humidity of 50%. Before printing the official samples, provisional trials varying nozzle temperature, nozzle size, printing speed and layer height were conducted to identify the printing parameters (Table 2) that result in the best printing quality.

### 2.6. Characterisation of 3D Printed Samples

#### 2.6.1. Tensile Testing

Tensile testing of the 3D printed samples was conducted using an Instron^®^ (model 5982) universal testing machine equipped with a 5 kN loadcell. All the samples were tested at a 2 mm/min cross-head displacement rate, and an extensometer of 10 mm was used to measure the tensile strain. The mechanical testing results were analysed in the statistical software Minitab^®^ 18 (Coventry, UK) using one-way analysis of variance (ANOVA) test. The significant differences among averages were calculated using Tukey’s method with 95% of confidence. After tensile testing, the fractured samples were analysed by scanning electron microscopy (SEM) in a Hitachi TM4000Plus SEM (Tokyo, Japan) operated at 5 kV using a secondary electrons detector.

#### 2.6.2. Fibre Alignment—X-ray Diffraction

In order to investigate fibre alignment during printing, 3D printed PLA/lyocell composite samples were analysed through XRD analysis. The 3D printed discs were analysed in a Panalytical Empyrean XRD (Worcestershire, UK) in transmission mode using CuKα radiation (40 kV; 40 mA) and a PixCel linear detector. First, the samples were analysed in a continuous 2θ scan mode between 5–45° 2θ using a scanning step of 0.01° and equivalent exposure time of 40 s at *φ* angles (disc rotation axis) of 0° and 90°. Then, a scan through the *φ* axis (0–180°) was conducted at the 2θ angle (approximately 21.2°) determined in the first scan using a step size of 0.5° and exposure time of 10 s.

#### 2.6.3. Dynamic Mechanical Analysis (DMA)

Dynamic thermomechanical analysis was conducted in a Perkin Elmer DMA8000 (Waltham, MA, USA) using single cantilever mode. A set of additional samples were heat-treated at 105 °C for 2 h to evaluate the influence of PLA crystallisation on the thermomechanical properties of the composites. All the samples were tested using a frequency of 1 Hz and strain of 0.05 from 25–140 °C. Two samples per condition were tested (neat PLA, L10% and L20%).

#### 2.6.4. Differential Scanning Calorimetry (DSC)

The 3D printed samples for DMA tests were characterised through DSC analysis. Small samples of 6–10 mg were extracted from the specimens and analysed in a Netzch DSC3500 Sirius (Selb, Germany) differential scanning calorimeter using aluminium crucibles from 20–200 °C at 10 °C/min with Nitrogen flow of 60 mL/min. The obtained scans were used to determine glass transition (*T_g_*), melting (*T_m_*), and cold crystallisation (*T_cc_*) temperatures. The PLA crystallinity of the samples was determined according to Equation 1 [47]:(1)$$X_{c}=\frac{\left(\Delta H_{m}-\Delta H_{cc}\right)}{\Delta H_{f}\times X_{PLA}}\times 100$$
where ∆*H_m_* and ∆*H_cc_* are the enthalpies of melting and cold crystallisation, respectively, ∆*H_f_* is the melting enthalpy of 100% crystalline PLA (93 J/g) [47], and *X_PLA_* is the weight fraction of PLA in the composite.

## 3. Results and Discussion

### 3.1. Filament Production and Characterisation

The lyocell fibres used in this study have a cellulose II structure with a degree of crystallinity of 76.7% (the XRD diffraction pattern of the fibres and corresponding Rietveld refinement are presented in Appendix A). Fibres used in thermoplastic-based composites undergo thermal cycles during processing, and therefore, knowing their thermal stability is necessary. The thermogravimetric (TG) curves of neat PLA and lyocell fibres and the corresponding derivative (DTG) are shown in Figure 2. As similarly reported in other studies [48], the lyocell fibres started to degrade rapidly above 316 °C, which is the onset temperature for the degradation reaction of cellulose II, and achieved the temperature of maximum degradation rate (*T_p_*) at 350 °C. The lyocell fibres have better thermal stability than other natural fibres used as reinforcement for thermoplastics, such as wood, hemp, and bamboo, which have an average *T_p_* of 292 °C and an average onset temperature of decomposition of 220 °C [49]. The Lyocell fibres only had a mass loss of 1.11 % up to 220 °C, which is advantageous as the typical processing temperature of PLA is between 180 and 200 °C).

The transverse and longitudinal sections of all the filaments analysed through optical microscopy are shown in Figure 3. Lower magnification images of all the filaments obtained by stereo microscopy can be seen in Appendix A. Most of the fibres are well distributed and partially aligned with the extrusion direction, which can be clearly observed in Figure 3a–d (indicated by white arrows), where the cross-section of the fibres is visible in the cross-section of the filament and the longitudinal section of the fibres are aligned with the extrusion direction. When higher fibre loading was used to produce the filaments, an increase in porosity was observed. The longitudinal section of the filaments with 10, 20, and 30% of fibres are also presented in Figure 3. Although optical microscopy is not the best method to calculate the porosity of composites, the porosity of the filament with 10% of fibres was estimated to be 0.48% (COV = 0.25), while the filaments with 20% and 30% of fibres had a porosity of 7.88% (COV = 0.10), and 5.57% (COV = 0.18), respectively.

As a reference, the micrographs of neat PLA filament (produced using the same conditions as the lyocell/PLA filaments) and a commercial PLA/wood filament are presented in Figure 4. First, it is possible to observe that there is no visual porosity for the neat PLA filament. On the other hand, the commercial PLA/wood filament has high porosity. This issue is not adequately discussed in the literature, and only a few works identify and comment on filament porosity. It is suggested that the causes for the high porosity content in these filaments may be related to residual moisture in the fibres, problems related to the extrusion parameters (feed rate and extrusion speed), and the long initial fibre length of the fibres used in this study [14]. Although not published yet, the authors have researched new processing routes to produce lyocell/PLA filaments using the same type and size of fibre, achieving filaments with improved properties. The compounding of lyocell fibres with PLA using a twin-screw extruder also affected the final length of the fibres in the composite, as demonstrated and discussed in the following section.

#### 3.1.1. Effect of Extrusion on Fibre Length

High shear forces are involved in twin-screw extruders, which are beneficial for compounding, but may also damage the fibres during processing. The morphology of the fibres extracted from the composites produced after each processing step is shown in Figure 5. The fibres were clearly affected by the extrusion process. The average fibre lengths were determined by optical microscopy for all the formulations after each processing step and are summarised in Table 3.

After undergoing extrusion twice, the fibres from all the composites had a similar length, having changed from 2.93 mm to approximately 0.159 ± 0.05 mm, which represents a length reduction of 94%. Although reduced fibre length may be beneficial for printability, it also affects the fibre/matrix load transfer and may lead to decreased mechanical properties. Virgin lyocell fibres have a very smooth surface which is detrimental for a good fibre/polymer stress transfer. The interfacial shear strength (IFSS) of lyocell fibres in PLA is reported to be approximately 10 MPa, which is lower than the IFSS of natural fibres such as flax and kenaf in PLA [50]. By using Kelly and Tyson equation [51], the critical fibre length calculated using Equation 2 for the fibres used in this study would be approximately 0.30 mm, considering a diameter of 11 μm and tensile strength of 536 MPa. Therefore, although the fibres still have an aspect ratio of >10 after processing, the lengths of the fibres are close to the critical limit for efficient reinforcement.
(2)$$L_{c}=\frac{\sigma_{f}\times d_{f}}{2\times \tau }$$
where, *L_c_* = Fibre critical length, *d_f_* = diameter of the fibre, and *τ* = IFSS.

#### 3.1.2. Tensile Properties

The tensile test results for the 3D printing filaments, i.e., ultimate tensile strength, Young’s modulus, and strain at break, are summarised in Table 4. Young’s modulus appears to have increased with the addition of fibres. However, although there was stress transfer between fibre and matrix, the short fibres and the presence of porosity of the samples (see Figure 3) acted as stress concentrators during tensile loading, especially in the filaments with higher fibre loading. Some of the fibres in the filament were not aligned with the extrusion direction and the aligned fibres were not enough to prevent failure at higher stresses due to their limited length, which affected especially the strain at break and tensile strength. However, all the filaments presented reasonably good properties in comparison to other studies reported in the literature (Appendix A). As a reference, a commercial PLA with 10 % of wood fibres was also tested. All the lyocell/PLA filaments had superior tensile strength and Young’s modulus.

### 3.2. 3D Printing

#### 3.2.1. Tensile Properties of 3D Printed Samples

Overall, all the filaments had good printability with no signs of nozzle blockage (clogging) during printing. Before printing the samples used for tensile testing, DMA, and XRD, the multiplier factor of the 3D printer software was adjusted to achieve acceptable surface and dimensional quality (as shown in Appendix A). Although the smooth surface of lyocell fibres and their short length after processing negatively affect the mechanical properties of the filaments, these characteristics are beneficial for printing, which enabled the use of small nozzles for printing. The filament with 10% of fibres could be printed with nozzles down to 0.25 mm with no apparent problems. However, in this work, we only report the mechanical test results of samples printed with a 0.5 mm nozzle, since with the smaller nozzle, clogging started to occur with the L20% and L30% formulations.

The resulting tensile properties of 3D printed samples are given in Table 5. Representative stress/strain curves of all the formulations can be visualized in Appendix A. The first thing to notice is that the ultimate tensile strength and Young’s modulus of the composites with 10% and 20% of fibres improved with the printing process. Since the samples were printed with a raster angle of 0°, which is parallel to the loading direction, this effect is attributed to increased fibre alignment caused by the shear stress during the printing process (better discussed in the next section) and possibly to PLA molecular alignment. The fibre alignment in fibre-reinforced composites plays an important role in fibre/matrix stress transfer and is a well-known method to improve mechanical properties, especially tensile strength [52].

The resulting fracture surfaces of 3D printed PLA and the L10% composite are presented in Figure 6. The fracture surfaces revealed a well-consolidated cross-section, with the presence of inter-bead porosity (voids between layers), that are usually observed in 3D printed samples by FDM [16,53]. In the L10% sample, it is possible to observe a combination of fibre pull-out and fibre breakage, indicated by the white and yellow arrows, respectively, in Figure 6. In fibres that are longer than the critical fibre length, the interfacial forces between fibre and matrix are high enough to prevent fibre pull-out, leading to fibre fracture. This load transfer between fibre and matrix up to fibre breakage improves Young’s modulus and tensile strength of the composite. On the other hand, part of the fibres will be shorter than the critical fibre length and will be prone to pull-out, especially if the smooth surface of the lyocell fibres is taken into consideration, which is translated to limited load transfer between fibre and matrix.

By comparing the tensile test results of the PLA filament and the 3D printed sample, it is possible to observe that the printing process resulted in high-quality samples, with minimal influence of printing defects on the tensile properties. The formulation containing 10 wt% fibres had the highest overall mechanical performance, achieving an ultimate tensile strength, Young’s modulus, and strain at break of 64.2 MPa, 4.56 GPa, and 4.93%, respectively. Although the tensile strength is slightly higher than neat PLA, the difference is not statistically relevant. The Young’s modulus, however, had a significant increase of 40% in comparison with neat PLA. Interestingly, the strain at break also increased with the addition of 10% of fibres and resulted in values with lower standard deviation. Tekinalp et al. (2019) reported similar tensile properties for their 3D printed samples using PLA reinforced with 10% of nanofibrillated cellulose (UTS ≈ 61–64 MPa; YM ≈ 4.2–4.8), but with strain at breaks below 2.5%, which is lower than that of the L10% composite [16].

The properties of the 3D printed lyocell/PLA composites are also superior to those of a commercial wood/PLA filament with 10% of fibres (values given in Table 5) printed with the same conditions. Nevertheless, the samples with 30% of fibres presented lower values of tensile strength, Young’s modulus and strain a break in comparison with the filament counterpart and 3D printed PLA. These results are attributed to the high porosity of the filaments and high-stress concentrations in regions outside the free-span which resulted in premature brittle failures. Therefore, only the L10% and L20% formulations were used for the DMA and XRD analyses presented in the next sections.

#### 3.2.2. Fibre Alignment in 3D Printed Samples

As discussed in the previous section, some of the 3D printed samples had better tensile properties than the filaments. This difference is attributed to the increased alignment of the fibres with the printing direction caused by the shear stresses involved during the printing process. This effect is clearly observed in Figure 7, where optical microscopy images of the filament and 3D printed sample of L10% formulation are presented. In the filament, the fibres are partially aligned with the extrusion direction, but due to the high-intensity mixing during the twin-screw extrusion, part of the fibres have a random orientation. However, when 3D printing is used, the shear stresses in the nozzle promote the alignment of the fibres in the printing direction. The shear rate in the nozzle during the printing process can be calculated according to Equations (S1)–(S4), shown in the Appendix A.

In order to evaluate and quantify the alignment of the fibres during the printing process, XRD analysis was conducted in transmission mode. By using this method, it is possible to have a comprehensive volumetric analysis of the samples, complementing traditional 2D image analysis. The Herman’s parameter and degree of ordering were calculated for samples 3D printed with different printing speeds, and hence extrusion shear, to evaluate if printing speed may improve the alignment of the fibres. A summary of the obtained results is given in Table 6. A complete description of the method and Equations used to calculate Herman’s parameter and degree of ordering is given in the Appendix A.

The degree of ordering and Herman’s parameter are used to describe how well-aligned cellulose crystallites are in a defined direction. High values of degree of ordering (π) (up to 100%) and Herman’s parameters (*f*) close to 1 indicate a maximum orientation, i.e., all the crystallites of the analysed phase are aligned. Low values of π and *f* = 0 indicate random orientation. From the values presented in Table 6, it is possible to observe that all the samples analysed had high values of π (average of 79%) and *f* above 0.6. These results corroborate the micrographs shown in Figure 6, where the fibres are aligned with the printing direction. However, the printing speed used in this study and the corresponding shear rate induced in the nozzle, does not seem to have a relevant influence on the alignment of the fibres, even with almost a three-fold difference. It is hypothesised that the dimensions of the fibres are sufficiently large for the shear rate induced by 0.5 mm nozzle to align the fibres during printing, even at relatively low printing speeds. The induced alignment of the fibres in the composites using FDM can be used to manufacture parts with controlled directional stiffening, which is an advantage in comparison with other manufacturing methods.

#### 3.2.3. Thermo-Mechanical Stability and DSC Analysis

Reinforcing PLA with fibres generally improves its thermo-mechanical properties, which can broaden its use in applications where thermo-mechanical stability is necessary [16,54,55,56]. The storage modulus (*E^’^*) and *tan δ* curves of neat PLA and composites with 10 and 20 wt% of fibres are presented in Figure 8a. The storage modulus curves have three distinct regions [54]. In the first one, between room temperature and 50 °C the materials are in a glassy state (below the *T_g_*) and are characterized by a linear plateau in this temperature range. In the transition region, between 50–70 °C, there is a sharp drop in the storage modulus, and the materials are transitioning from a glassy state to a rubbery state. Above approximately 70 °C the PLA and the composites are characterised by a rubbery behaviour with minimum storage modulus. *Tan δ* is the ratio between loss modulus (*E’*’) and storage modulus (*E’*) and is used to identify the glassy to rubbery transition. The temperature with the highest tan δ value is often stated as the glass transition (*T_g_*) temperature. When the lyocell fibres were added, the intensity of tan δ peak was considerably reduced, which indicates less viscous behaviour and better thermomechanical stability. In addition, above approximately 90 °C, it is possible to observe a slight increase in the storage modulus in all the samples (see insert of Figure 8a) related to the beginning of crystalisation of PLA. This effect starts at lower temperaure in the composites, which demonstrates that the addition of the fibres facilitates PLA crystallisation.

The presence of crystalline regions decreases the mobility of PLA molecules. Therefore, an increase in PLA crystallinity is translated into higher storage modulus at higher temperatures improving thermal-mechanical stability. Additional samples were heat-treated at 105 °C for 2 h to evaluate the effect of the increase in PLA crystallinity in the thermo-mechanical behaviour of the 3D printed samples. The samples with 10 and 20 wt% of fibres did not have dimensions affected (below 1%) after heat treatment. The 3D printed PLA sample, on the other hand, had a decrease of 13 % in length and an increase of 5% and 13% in width and thickness, respectively (see Appendix A). This effect is expected in some grades of PLA and is more evident and anisotropic in 3D printed samples.

The storage modulus and tan δ curves of the heat-treated samples are presented in Figure 8b. The heat-treated samples had a considerable improvement in the thermo-mechanical stability, characterised by the lower values of *tan δ* and higher storage modulus above the transition temperature. The heat-treated neat PLA sample was tested only as a reference, but it is worth mentioning that this type of treatment would not be possible for 3D printed objects using this unreinforced PLA grade since the anisotropic shrinkage/swelling caused by the increase in crystallinity would cause considerable changes in the object’s dimensions. This is considered a major advantage of using fibres in PLA, enabling post-printing heat treatment to increase PLA crystallinity while preserving 3D object shape and dimensions.

The overall properties obtained from the DMA tests of all the specimens (as printed and heat-treated) are summarised in Table 7. First, there is a continuous decrease in the maximum *tan δ* values with the addition of the fibres. In the heat-treated samples, a similar trend is observed, but with lower absolute values. The maximum *tan δ* temperature and loss modulus (*E’’*) peak temperature were not affected by the addition of the fibres, around 67 and 61 °C, respectively. Nevertheless, when heat-treated all the samples had an increase in the *tan δ* and *E’’* peak temperatures, 72 and 65 °C, respectively. The storage modulus (*E’*) at different temperatures is also given in Table 7. At 30 °C, the composite samples have a gradual increase in E’ with the addition of fibres and are characterised by a more pronounced difference between the as-printed and heat-treated samples. This effect is more evident in the values of E’ at higher temperatures. At 80 °C, the *E’* of heat-treated neat PLA is 22.5× higher than the sample without treatment, while for the L10% and L20% this difference is 64.7× and 40×, respectively. Additionally, the *E’* at 80 °C of heat-treated L10% and L20% samples are 60.6× and 72.5× higher than the *E’* of neat PLA, respectively. It was demonstrated that the thermo-mechanical stability of the composites with lyocell fibres could be drastically improved by post-printing heat-treatment without any hurdles associated with dimensional changes.

The 3D printed samples used for the DMA tests were also analysed by DSC to verify the crystallinity of PLA and thermophysical properties of the composites and complement the information given in Table 7. Representative DSC curves during heating of the as-printed and heat-treated samples are presented in Figure 9. In Figure 9a it is possible to observe that all the samples have a typical DSC curve for the 2003D PLA grade, with an endothermic *Tg* peak at 62–64 °C, exothermic *T_cc_* (cold crystallisation temperature) peak between 114–123 °C and endothermic Tm peak between 150–153 °C. In contrast, with heat treatment, only the Tm peaks can be observed. Since most of the available PLA chains are already crystalline, there are no signs of *T_cc_* in these samples.

A summary of all the results is shown in Table 8. First of all, as observed in the DMA results, the *T_cc_* temperature is lower for the composite materials which is an indication of the influence of the fibres on the crystallisation behaviour of PLA. However, after 2 h at 105 °C all the samples had similar PLA crystallinity, between 28–29%. It is worth mentioning that this PLA grade (2003D) has higher D-isomer content than other PLA grades and as an injection moulding grade, it is more difficult to crystallise than other grades [57]. In fact, during cooling at 10 °C/min (same rate used for heating in the DSC analysis), this grade does not have any signs of crystallisation. The endothermic *T_g_* peaks had little influence by the addition of fibres and although the *T_g_* peak of the composites is slightly lower than neat PLA, the starting temperature of the transition is more or less the same for all the samples. The Tg temperatures obtained by DSC are in agreement with the DMA results and are between the *tan δ* and *E’’* values (shown in Table 7). In addition, the heat-treated L10% and L20% composites present a bimodal melting endotherm (Figure 9b) that is attributed to the melting of the metastable α form of PLA (α’) potentially formed during 3D printing [53,57,58,59,60]. The first peak of fusion is related to the simultaneous melting of the primary α crystals formed during printing and recrystallisation of α’ to α crystal form. The second peak of fusion is attributed to the melting of α crystal form generated in the recrystallisation process [61].

## 4. Conclusions

Composite filaments of PLA reinforced with lyocell fibres were successfully produced for fused deposition modelling and were easily 3D printed into testing samples. However, the use of twin-screw extrusion damages the fibres during processing, considerably reducing the fibre length. In addition, increasing the fibre content using this process caused excessive porosity in the filaments. Nevertheless, the filaments had an increased Young’s modulus and could be easily 3D-printed with a relatively small nozzle diameter of 0.5 mm. The 3D printing process induced fibre alignment, and this is thought to account for the tensile strength and Young’s modulus improvement when 3D-printed specimens were compared with the filaments. Three-dimensionally-printed specimens with 10% of lyocell fibres had an average ultimate tensile strength, Young’s modulus, and strain at break of 64.2 MPa, 4.56 GPa, and 4.93 %, respectively. The addition of lyocell fibres also improved the storage modulus of PLA and reduced the maximum tan δ associated with the glassy/rubbery transition. In addition, by heat-treating the composite samples in a temperature range close to *T_cc_*, the crystallinity of PLA could be increased without dimensional changes. This increase in crystallinity was translated into composites with outstanding thermo-mechanical stability, with storage moduli at 80 °C 61 to 72 times higher than that of neat PLA.

The use of 3D printing to fabricate parts made of polymers reinforced with short lyocell fibres can be explored as a way to selectively improve the mechanical (controlled directional stiffening) and thermo-mechanical stability of objects, extending the use of PLA to load-bearing structures with complex geometry that can be used, for example, from tailor-made furniture or building components to biomedical and dentistry applications.

## Figures and Tables

**Figure 1 polymers-14-03991-f001:**
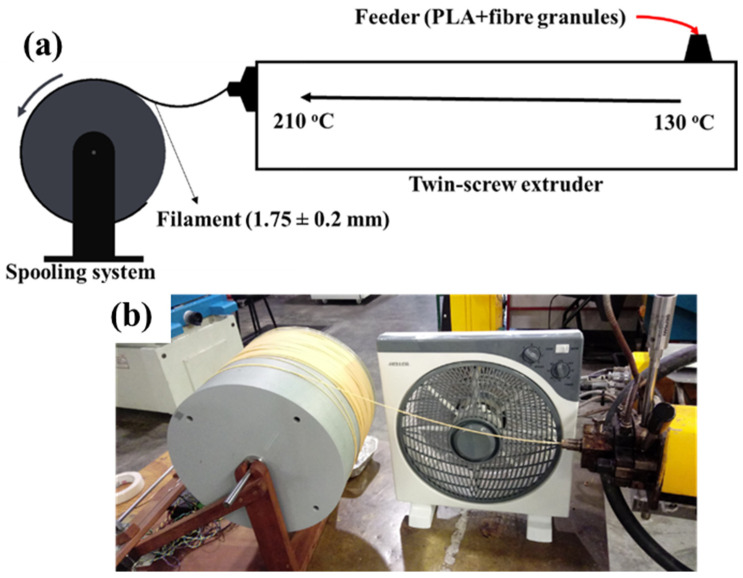
Schematic representation of the filament production process (**a**) and picture of the spooling system (**b**).

**Figure 2 polymers-14-03991-f002:**
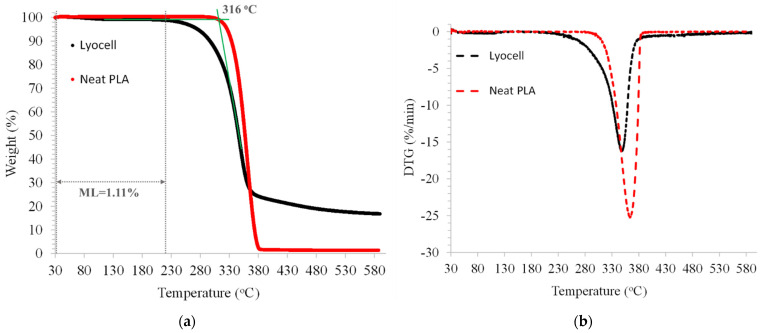
TG (**a**) and the corresponding derivative DTG (**b**) curves of neat PLA and lyocell fibres.

**Figure 3 polymers-14-03991-f003:**
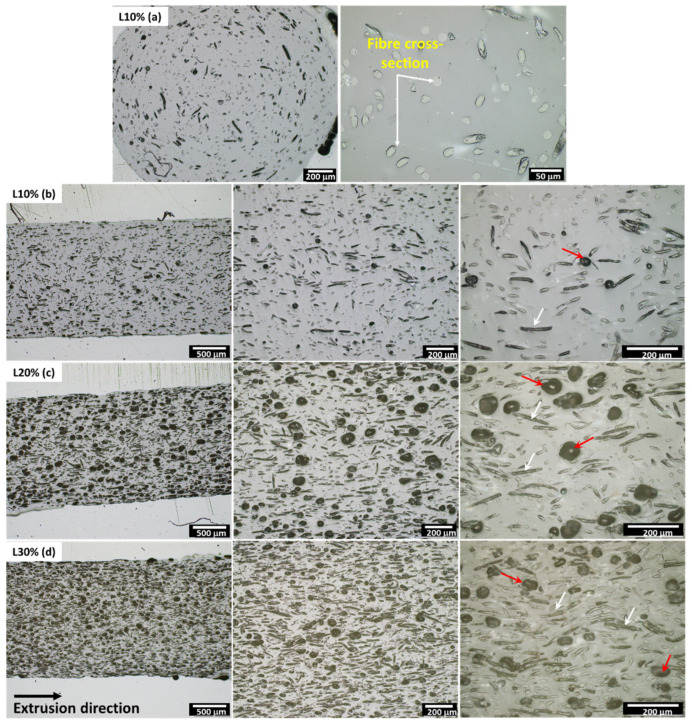
Optical microscopy images of L10% filament transverse section (**a**) and sections parallel to the extrusion direction of the L10% (**b**), L20% (**c**), and L30% (**d**) filaments. The red arrows indicate the pores and the white arrows indicate the fibres parallel to the extrusion direction.

**Figure 4 polymers-14-03991-f004:**
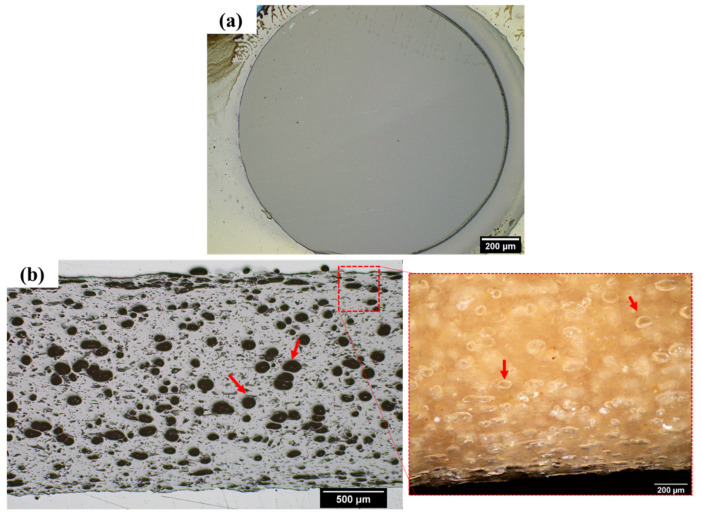
Optical microscopy images of filaments of neat PLA (**a**) and commercial wood-filled PLA (**b**).

**Figure 5 polymers-14-03991-f005:**
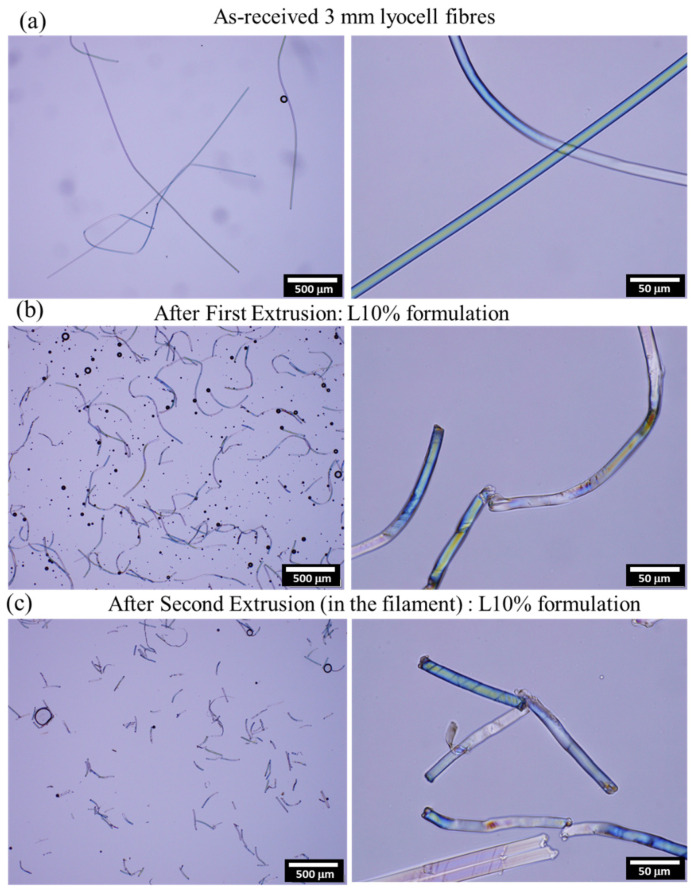
Optical microscopy images of the as-received fibres (**a**) and fibres extracted from the composites after the first (**b**) and second extrusion (**c**).

**Figure 6 polymers-14-03991-f006:**
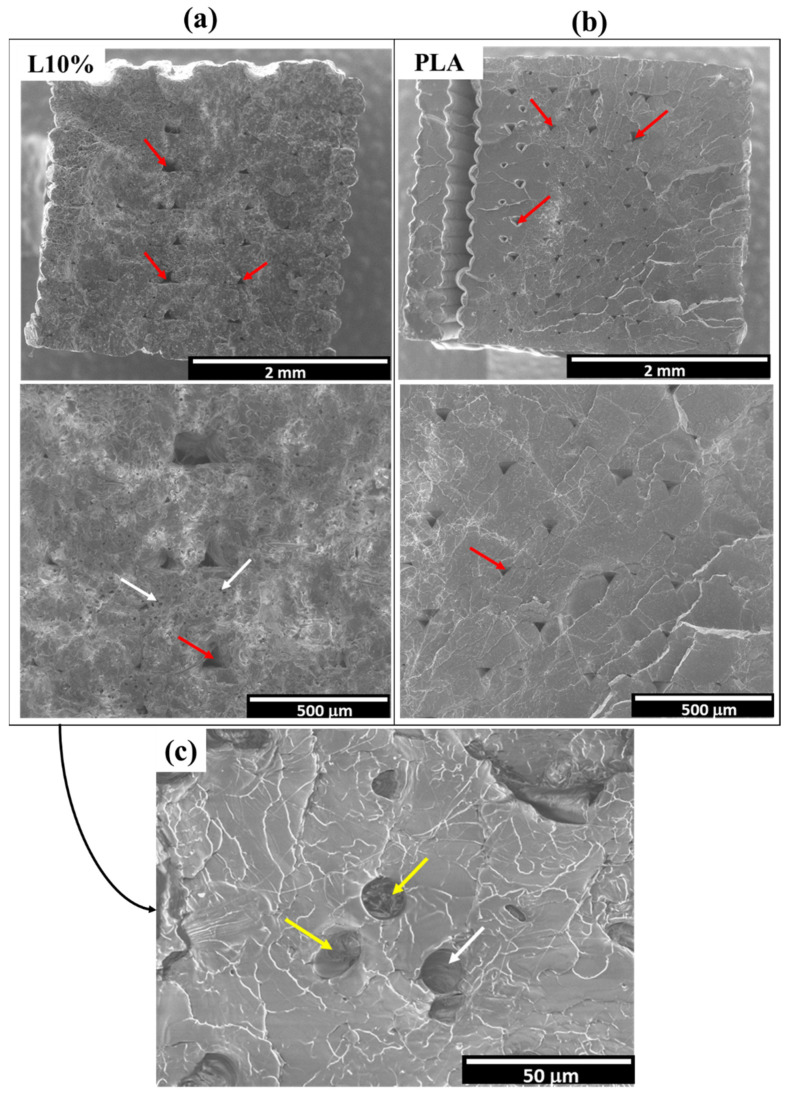
SEM images of the fracture surfaces of tested L10% (**a**) and PLA (**b**) 3D printed samples. The red arrows indicate inter-bead porosity (**a**,**b**) and the white and yellow arrows indicate fibre pull-out and fibre breakage in the L10% sample (**c**).

**Figure 7 polymers-14-03991-f007:**
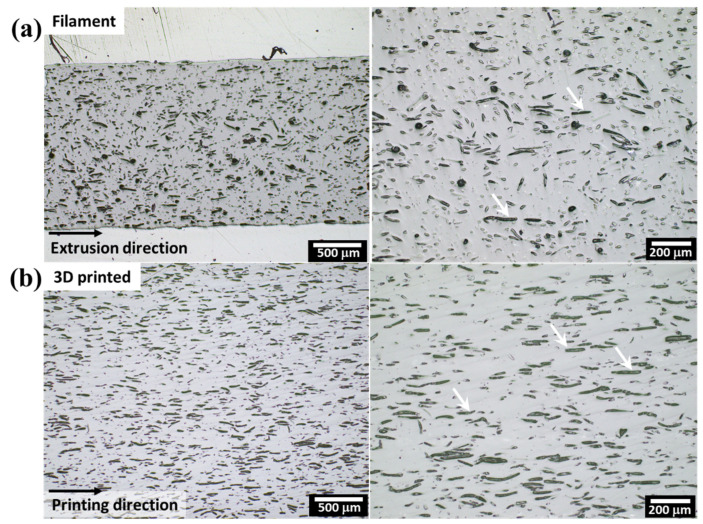
Optical microscopy images of filament (**a**) and 3D printed sample (**b**) of composite with 10% of lyocell fibres. The white arrows indicate the longitudinal section of the fibres aligned with the extrusion/printing direction.

**Figure 8 polymers-14-03991-f008:**
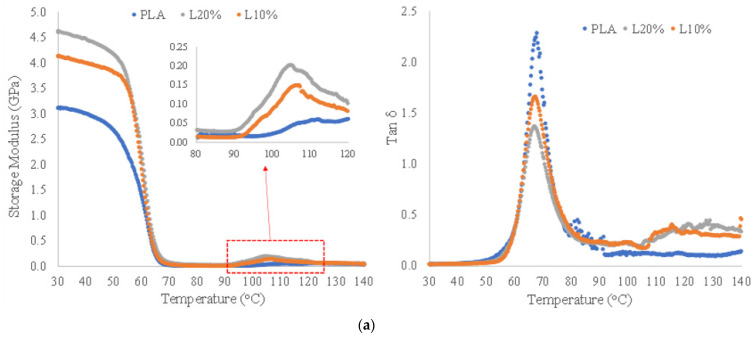
Storage modulus and tan δ obtained from DMA tests of 3D printed samples of PLA, L10%, and 20% (**a**) and heat-treated samples at 105 °C for 2 h (**b**).

**Figure 9 polymers-14-03991-f009:**
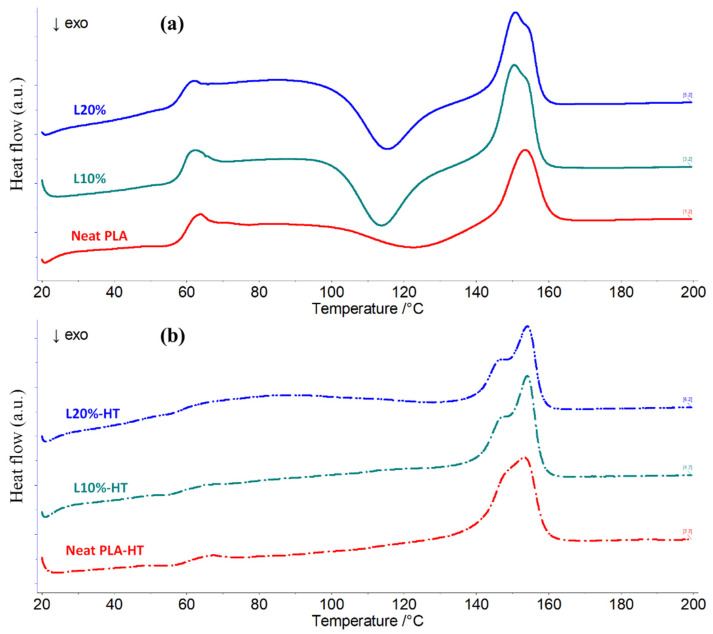
DSC curves during heating of 3D printed samples of as-printed samples (**a**), PLA, L10%, and L20%, and heat-treated samples (**b**) used for DMA analysis.

**Table 1 polymers-14-03991-t001:** Composition and extrusion conditions (for constant die pressure and uniform filament extrusion) used in the production of lyocell/PLA filaments.

ID	Fibre Content (wt%)	PLA Content (wt%)	Screw Speed (rpm)	Feed Rate (a.u.)
Neat PLA	0	100	18	4.4
L10%	10	90	30–36	6.1
L20%	20	80	35	5.5
L30%	30	70	20–25	7.4

**Table 2 polymers-14-03991-t002:** Parameters used for the 3D printed samples.

Parameter	Value
Infill density	100%
Nozzle diameter	0.5 mm
Raster angle	0° (for all the layers)
Bed temperature	70 °C
Nozzle temperature	215 °C
Printing speed	1800 mm/min
Layer height	0.1 mm

**Table 3 polymers-14-03991-t003:** Average fibre length with the corresponding coefficient of variation (COV) of the composites after the first extrusion and in the final filaments.

Condition	Processing	Fibre Length (μm)		
		*n*	Average	*COV*
As-received fibres		17	2930	0.04
L10%	1st Extrusion	106	570.8	0.71
	Filament	100	157.9	0.25
L20%	1st Extrusion	100	925.7	0.63
	Filament	105	157.3	0.33
L30%	1st Extrusion	102	722.9	0.53
	Filament	107	161.9	0.29

**Table 4 polymers-14-03991-t004:** Tensile test results of PLA/lyocell filament samples. COV is presented in parentheses. Same letters (a, b or c) indicate that there is no statistical difference between formulations.

Formulation	Tensile Properties of Filaments (*n* = 6)			
	Filament Diameter (mm)	*UTS* (MPa)	*E* (GPa)	*ε_break_* (%)
Neat PLA	1.76 (0.01)	59.1 ^a^ (0.01)	3.41 ^d^ (0.02)	4.89 ^a^ (0.16)
L10%	1.72 (0.01)	57.4 ^a,b^ (0.01)	3.91 ^c^ (0.03)	2.44 ^b^ (0.12)
L20%	1.70 (0.02)	54.8 ^c^ (0.01)	4.23 ^b^ (0.02)	2.45 ^b^ (0.12)
L30%	1.70 (0.02)	55.8 ^b,c^ (0.04)	5.05 ^a^ (0.06)	1.94 ^b^ (0.11)
Commercial wood filament *	1.71 (0.01)	52.9 (0.01)	3.59 (0.06)	2.43 (0.11)

UTS—Tensile strength; E—Young’s modulus; ε_break_—Strain at break. *—Used as a reference: Commercial PLA filament with 10% of wood fibres (Wood-Filled PLA 1.75 mm Dark Brown, Imagin Plastics Ltd., Auckland, New Zealand).

**Table 5 polymers-14-03991-t005:** Tensile test results of PLA/lyocell 3D printed samples. The coefficient of variation (COV) is presented in parentheses. Same letters (a, b or c) indicate that there is no statistical difference between formulations.

Formulation	Tensile Properties of 3D Printed Samples (*n* = 3)		
	*UTS* (MPa)	*E* (GPa)	ε_break_ (%)
Neat PLA	62.8 ^a^ (0.01)	3.26 ^b^ (0.03)	3.61 ^b^ (0.21)
L10%	64.2 ^a^ (0.01)	4.56 ^a^ (0.03)	4.93 ^a^ (0.03)
L20%	57.1 ^b^ (0.02)	4.58 ^a^ (0.03)	1.99 ^c^ (0.17)
L30%	47.5 ^c^ (0.01)	4.90 ^a^ (0.03)	1.58 ^c^ (0.09)
Commercial wood filament *	57.6 (0.01)	3.53 (0.05)	2.54 (0.07)

UTS—Tensile strength; E—Young’s modulus; ε_break_—Strain at break. *—Used as a reference: Commercial PLA filament with 10% of wood fibres (Wood-Filled PLA 1.75 mm Dark Brown, Imagin Plastics Ltd., Auckland, New Zealand).

**Table 6 polymers-14-03991-t006:** Alignment parameters obtained by XRD analysis of samples 3D printed with different printing conditions.

Condition	Printing Speed (mm/min)	Shear Rate $\left(\dot{\gamma )}\;(s^{-1}\right)$	Degree of Ordering (π) (%)	Herman’s Parameter (*f*)	Peak Position (φ)	FWHM (o)
L10%	1800	122	78.9	0.661	89.77	37.96
	4800	326	78.8	0.602	90.39	38.08
L20%	1800	122	78.1	0.587	89.36	39.34
	4800	326	79.2	0.635	89.77	37.50

**Table 7 polymers-14-03991-t007:** Summary of DMA results of as printed and heat-treated samples. The values are the average of two tested samples per condition.

Condition	Heat Treatment	Max. *tan δ*	Max. *tan* *δ* Temp. (°C)	*E’’* Peak Temp. (°C)	*E’* at 30 °C (GPa)	*E’* at 60 °C (GPa)	*E’* at 80 °C (GPa)
Neat PLA	-	2.25	67.40	61.60	3.08	1.32	0.016
	105 °C—2 h	0.31	71.60	65.00	3.17	2.38	0.36
PLA-L10%	-	1.69	67.10	61.40	3.98	1.96	0.015
	105 °C—2 h	0.20	71.30	65.60	4.32	3.40	0.97
PLA-L20%	-	1.38	67.00	61.20	4.19	2.00	0.029
	105 °C—2 h	0.19	74.00	65.90	4.43	3.54	1.16

**Table 8 polymers-14-03991-t008:** Summary of DSC results of the samples used in the DMA analysis.

Condition	Heat Treatment	*T_g_* (°C)	*T_cc_* Peak (°C)	∆*H_cc_* (J/g)	∆*H_m_* (J/g)	*T_m_* Peak (°C)	*X_c_* (%)
Neat PLA	-	63.8	122.7	−10.56	15.72	153.5	5.54
	105 °C—2 h	-	-	-	26.80	153.2	28.60
PLA-L10%	-	62.4	113.9	−20.9	24.14	150.5	3.87
	105 °C—2 h	-	-		24.76	150.1	29.36
PLA-L20%	-	62.1	115.5	−18.27	20.59	150.8	3.12
	105 °C—2 h	-	-	-	20.9	150.5	28.08

*T_cc_*—Cold crystallisation temperature; *T_g_*—Glass transition temperature; *T_m_*—Melting temperature; ∆*H_cc_*—Cold crystallisation heat; ∆*H_m_*—Heat of melting; *X_c_*—PLA crystallinity.

## Data Availability

Additional data and materials can be provided upon request.

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
