# Peer review of "Production and Assessment of Poly(Lactic Acid) Matrix Composites Reinforced with Regenerated Cellulose Fibres for Fused Deposition Modelling"

_polymers, 2022, doi:10.3390/polym14193991_

Round 1
Reviewer 1 Report
The manuscript titled “Production and assessment of polylactic acid matrix composites reinforced with regenerated cellulose fibres for fused deposition modelling” by Gauss, C.; et al. is an original scientific work where the authors analyze the impact of cellulose fibre content on the mechanical performance of composites made by polylactic acid (PLA). Multitude of complementary techniques are used to fully characterize these cellulose fibre-reinforced PLA composites. The comparative analysis conducted by the authors shows the best mechanical performance found was for those blends of PLA with 10 wt% of cellulose fibres. The conducted research is highly innovative combining the above described approaches and could have a strong relevance on the composite industry. Moreover, the present methodology can be fully implemented to other matrixes and/or fillers which could have positive impact on society. The gathered findings may be relevant for the examined field. The results achieved are well-discussed during the main body of the reported manuscript. The scientific paper is well written. In my opinion the present manuscript is innovative and the methodological approached used matches with the scope of Polymers journal. For the above described reasons, I recommend the publication in Polymers once the following remarks will be fixed:
--------
KEYWORDS
The keywords used by the authors are accurate and well-chosen. I may also introduce the following terms “cellulose” “polylactic acid or PLA” and “3D printing technology”.
--------
INTRODUCTION
“(…) three-dimensional (3D) printing, is a collection (…). One of the most common (…) is fused deposition modelling (FDM), also known as fused filament fabrication (FFF) [3].” (lines 25-29). Authors should also state other existing 3D printing technologies like electrohydrodynamic (EHC) [1], laser-assisted [2] or based on acoustic [3] modes.
[1] Jiang, C.; et al. Quantitative Investigation of the Process Parameters of Electrohydrodynamic Direct-Writing and Their Effects on Fiber Surface Roughness and Cell Adhesion. Polymers 2020, 12, 2475. https://doi.org/10.3390/polym12112475.
[2] Borg, G.; et al. Filament Development for Laser Assisted FFF 3D Printing. J. Manuf. Mater. Process 2021, 5, 115. https://doi.org/10.3390/jmmp5040115.
[3] Habibi, M.; et al. Direct sound printing. Nat. Commun. 2022, 13, 1800. https://doi.org/10.1038/s41467-022-29395-1.
“Some of the printing parameters (…) feed rate and raster angle.” (lines 59-61). Authors highlight the parameter conditions necessary to be taken into account. It may be interesting also to remark potential drawbacks/limitations linked to layer-by-layer 3D printing technologies. In this framework, I encourage the authors to state nozzle clogging [4] and satellite droplets [5], respectively and how these detrimental aspects have been overcome (or minimized) in the current study. This last information should be also indicated in the respective Materials & Methods section (2.5. 3D printing, from line 155).
[4] Gutiérrez, E.; et al. Decrease of Nozzle Clogging through Fluid Flow Control. Metals 2020, 10, 1420. https://doi.org/10.3390/met10111420.
[5] Oktavianty, O.; et al. Enhancing Droplet Quality of Edible Ink in Single and Multi-Drop Methods by Optimization the Waveform Design of DoD Inkjet Printer. Processes 2022, 10, 91. https://doi.org/10.3390/pr10010091.
“The combination of good mechanical properties, (…) to other bio-based fibres.” (lines 74-76). Here (or in the Conclusions section, from line 558), it may be a perfect opportunity to introduce potential industrial applications (food packaging, binders, scaffolds for dentistry components, building manufacturing, biomedical targeted on tissue engineering, ...) that may lead the manufacturing of next-generation of sustainable products that may positively impact on the human welfare.
--------
MATERIALS AND METHODS
2.6.4. Differential Scanning Calorimetry (DSC). “(…) according to Equation 1 [44]” (line 209). Why did the authors enter this eqn. 1 in the Materials and Methods and the eqn. 2 (line 296) in the Results and Discussion section? I may homogenize this aspect. Moreover, authors should fix the following flaw “(3)” (line 211) being exchanged by “(1)”.
--------
RESULTS AND DISCUSSION
In general terms, the results are displayed clearly for the potential readers. Nevertheless, the following aspects must be further addressed:
I) Authors should add a schematic representation according the composite fabrication.
II) Table 5 (line 352). The Young’s modulus values from L10% and L20% formulation samples are very similar (4.56 GPa vs 4.58 GPa, respectively). Could the authors carry out some further statistical analysis like Student’s test or analysis of variance (ANOVA) to discern the statistical significance of the aforementioned results?
III) What is the Poisson’s ratio value used by the authors to calculate the mechanical properties of the tested composites? This information should be included here or in the respective Materials & Methods section.
IV) Figure 5 (line 383) in addition to “The resulting fracture surfaces (…) In the L10% sample, it is possible to observe a combination of fibre pull-out and fibre breakage” (lines 342-346). Authors attempt to explain the observation of fractures on the fabricated composites is based on the porosity induced by moisture trapped inside the composite body (lines 251-259). Even if I am agree with this hypothesis, authors should not to discharge the possibility the fracture phenomena was caused by the high-interfacial forces exerted between the PLA matrix and the cellulose fillers. Recently, it has been reported an interesting work where functionalized surfaces with pure cellulose nanocrystals [6] are faced to composites made by matrices with polar nature and cellulose rich-content hemp/flax fibres [7]. This study concluded that the breakage of bundles and the single hemp/flax fibres was undertaken due to these high-interfacial forces. Authors should add a statement to complete the discussion on this regard.
[6] Marcuello, C.; et al. Langmuir-Blodgett Procedure to Precisely Control the Coverage of Functionalized AFM Cantilevers for SMFS Measurements: Application with Cellulose Nanocrystals. Langmuir 2018, 34, 9376-9386. https://doi.org/10.1021/acs.langmuir.8b01892.
[7] Berzin, F.; et al. Influence of the polarity of the matrix on the breakage mechanisms of Lignocellulosic fibers during twin-screw extrusion. Polym. Compos. 2020, 41, 1106-1117. https://doi.org/10.1002/pc.25442.
--------
CONCLUSIONS
This section is clear and concise. Authors well-discussed the main outcomes found in the present manuscript. As aforementioned, authors should indicate some potential Industrial applications (please, see comments poured in Introduction section).
--------
BIBLIOGRAPHY
The reference style is in the proper format of Polymers journal. No further actions are required in this section.
--------
OVERVIEW AND FINAL COMMENTS
The submitted work is well-designed and the gathered results are interesting for the design and creation of more durable and sustainable composites with improved mechanical performance. For this reason, I will recommend the present scientific manuscript for further publication in Polymers once all the aforementioned suggestions will be properly fixed.
Author Response
Reviewer 1:
Comments and Suggestions for Authors
The manuscript titled “Production and assessment of polylactic acid matrix composites reinforced with regenerated cellulose fibres for fused deposition modelling” by Gauss, C.; et al. is an original scientific work where the authors analyze the impact of cellulose fibre content on the mechanical performance of composites made by polylactic acid (PLA). Multitude of complementary techniques are used to fully characterize these cellulose fibre-reinforced PLA composites. The comparative analysis conducted by the authors shows the best mechanical performance found was for those blends of PLA with 10 wt% of cellulose fibres. The conducted research is highly innovative combining the above described approaches and could have a strong relevance on the composite industry. Moreover, the present methodology can be fully implemented to other matrixes and/or fillers which could have positive impact on society. The gathered findings may be relevant for the examined field. The results achieved are well-discussed during the main body of the reported manuscript. The scientific paper is well written. In my opinion the present manuscript is innovative and the methodological approached used matches with the scope of Polymers journal. For the above described reasons, I recommend the publication in Polymers once the following remarks will be fixed:
Answer:
Thank you. Your revision will help us improve the quality of our manuscript.
--------
KEYWORDS
The keywords used by the authors are accurate and well-chosen. I may also introduce the following terms “cellulose” “polylactic acid or PLA” and “3D printing technology”.
--------
INTRODUCTION
“(…) three-dimensional (3D) printing, is a collection (…). One of the most common (…) is fused deposition modelling (FDM), also known as fused filament fabrication (FFF) [3].” (lines 25-29). Authors should also state other existing 3D printing technologies like electrohydrodynamic (EHC) [1], laser-assisted [2] or based on acoustic [3] modes.
[1] Jiang, C.; et al. Quantitative Investigation of the Process Parameters of Electrohydrodynamic Direct-Writing and Their Effects on Fiber Surface Roughness and Cell Adhesion. Polymers 2020, 12, 2475. https://doi.org/10.3390/polym12112475.
[2] Borg, G.; et al. Filament Development for Laser Assisted FFF 3D Printing. J. Manuf. Mater. Process 2021, 5, 115. https://doi.org/10.3390/jmmp5040115.
[3] Habibi, M.; et al. Direct sound printing. Nat. Commun. 2022, 13, 1800. https://doi.org/10.1038/s41467-022-29395-1.
Answer:
Thank you for your suggestion. We have modified this section including the main printing methods according to ASTM standard F2792. The methodologies from the above references are not common and we think that they would not be relevant to this work.
“Some of the printing parameters (…) feed rate and raster angle.” (lines 59-61). Authors highlight the parameter conditions necessary to be taken into account. It may be interesting also to remark potential drawbacks/limitations linked to layer-by-layer 3D printing technologies. In this framework, I encourage the authors to state nozzle clogging [4] and satellite droplets [5], respectively and how these detrimental aspects have been overcome (or minimized) in the current study. This last information should be also indicated in the respective Materials & Methods section (2.5. 3D printing, from line 155).
[4] Gutiérrez, E.; et al. Decrease of Nozzle Clogging through Fluid Flow Control. Metals 2020, 10, 1420. https://doi.org/10.3390/met10111420.
[5] Oktavianty, O.; et al. Enhancing Droplet Quality of Edible Ink in Single and Multi-Drop Methods by Optimization the Waveform Design of DoD Inkjet Printer. Processes 2022, 10, 91. https://doi.org/10.3390/pr10010091.
Answer:
Thank you for your suggestion. This section has been modified to highlight the problems associated with FDM. “When fibre-reinforced composites are used, especially with high fibre content (> 20 wt%), non-uniform printing, nozzle blockage (clogging), and poor interlayer adhesion are generally reported, mainly due to the highly viscous nature of the composites [23,31,32]. There-fore, modification of the printing parameters is often necessary to achieve acceptable printing quality.”
In addition the following sentence was included in materials and methods: “Before printing the official samples, provisional trials varying nozzle temperature, nozzle size, printing speed and layer height were conducted to identify the printing parameters (Table 2) that result in the best printing quality.”
“The combination of good mechanical properties, (…) to other bio-based fibres.” (lines 74-76). Here (or in the Conclusions section, from line 558), it may be a perfect opportunity to introduce potential industrial applications (food packaging, binders, scaffolds for dentistry components, building manufacturing, biomedical targeted on tissue engineering, ...) that may lead the manufacturing of next-generation of sustainable products that may positively impact on the human welfare.
Answer:
The conclusion was modified accordingly: “The use of 3D printing to fabricate parts made of polymers reinforced with short lyocell fibres can be explored as a way to selectively improve the mechanical (controlled directional stiffening ) and thermo-mechanical stability of objects, extending the use of PLA to load-bearing structures with complex geometry that can be used, for example, from tailor-made furniture or building components to biomedical and dentistry applications.”
MATERIALS AND METHODS
2.6.4. Differential Scanning Calorimetry (DSC). “(…) according to Equation 1 [44]” (line 209). Why did the authors enter this eqn. 1 in the Materials and Methods and the eqn. 2 (line 296) in the Results and Discussion section? I may homogenize this aspect. Moreover, authors should fix the following flaw “(3)” (line 211) being exchanged by “(1)”.
Answer:
Corrected. However, the Equation 2 is presented in the results and discussion because it is only part of the discussion to explain the relevance of fibre length on polymer reinforcement. We think it would be better placed there and not in materials and methods.
RESULTS AND DISCUSSION
In general terms, the results are displayed clearly for the potential readers. Nevertheless, the following aspects must be further addressed:
- I)Authors should add a schematic representation according the composite fabrication.
- II)Table 5 (line 352). The Young’s modulus values from L10% and L20% formulation samples are very similar (4.56 GPa vs 4.58 GPa, respectively). Could the authors carry out some further statistical analysis like Student’s test or analysis of variance (ANOVA) to discern the statistical significance of the aforementioned results?
III) What is the Poisson’s ratio value used by the authors to calculate the mechanical properties of the tested composites? This information should be included here or in the respective Materials & Methods section.
Answer:
- Thank you for the suggestion. We have included a schematic representation for the production of the filaments (Figure 1).
- Yes, the values are very similar and are statistically the same. All the tables reporting the mechanical properties have the following statement: “Same letters (a, b or c) indicate that there is no statistical difference between formulations.” In this case, both conditions have the same letter (a) and therefore, can be considered indifferent using the ANOVA test. From the materials and methods section: “The mechanical testing results were analyzed in the statistical software Minitab® 18 using one-way analysis of variance (ANOVA) test. The significant differences among averages were calculated using Tukey's method with a 95% of confidence “
- The Poisson’s ratio is not necessary to determine the tensile properties we have reported. The elongation (in the direction of the applied load) was obtained by using a contact extensometer in the middle of the free-span of the sample. Therefore the Young’s modulus can be calculated as the slope of the linear region of the stress/strain curve obtained during testing (as per ASTM D638).
- IV)Figure 5 (line 383) in addition to “The resulting fracture surfaces (…) In the L10% sample, it is possible to observe a combination of fibre pull-out and fibre breakage” (lines 342-346). Authors attempt to explain the observation of fractures on the fabricated composites is based on the porosity induced by moisture trapped inside the composite body (lines 251-259). Even if I am agree with this hypothesis, authors should not to discharge the possibility the fracture phenomena was caused by the high-interfacial forces exerted between the PLA matrix and the cellulose fillers. Recently, it has been reported an interesting work where functionalized surfaces with pure cellulose nanocrystals [6] are faced to composites made by matrices with polar nature and cellulose rich-content hemp/flax fibres [7]. This study concluded that the breakage of bundles and the single hemp/flax fibres was undertaken due to these high-interfacial forces. Authors should add a statement to complete the discussion on this regard.
[6] Marcuello, C.; et al. Langmuir-Blodgett Procedure to Precisely Control the Coverage of Functionalized AFM Cantilevers for SMFS Measurements: Application with Cellulose Nanocrystals. Langmuir 2018, 34, 9376-9386. https://doi.org/10.1021/acs.langmuir.8b01892.
[7] Berzin, F.; et al. Influence of the polarity of the matrix on the breakage mechanisms of Lignocellulosic fibers during twin-screw extrusion. Polym. Compos. 2020, 41, 1106-1117. https://doi.org/10.1002/pc.25442.
Answer:
Thank you for your suggestion. The mechanisms involved and analyses used in the mentioned paper are different then the ones used in our study. However, we agree that additional information on the fracture behaviour would be important. We have modified this section and included the following sentence:
“In fibres that are longer than the critical fibre length, the interfacial forces between fibre and matrix are high enough to prevent fibre pull-out, leading to fibre fracture. This load transfer between fibre and matrix up to fibre breakage improves Young’s modulus and tensile strength of the composite. On the other hand, part of the fibres will be shorter than the critical fibre length and will be prone to pull-out, especially if the smooth surface of the lyocell fibres is taken into consideration, which is translated to limited load transfer between fibre and matrix.”
CONCLUSIONS
This section is clear and concise. Authors well-discussed the main outcomes found in the present manuscript. As aforementioned, authors should indicate some potential Industrial applications (please, see comments poured in Introduction section).
Answer:
Thanks. The conclusion has been modified accordingly.
--------
BIBLIOGRAPHY
The reference style is in the proper format of Polymers journal. No further actions are required in this section.
--------
OVERVIEW AND FINAL COMMENTS
The submitted work is well-designed and the gathered results are interesting for the design and creation of more durable and sustainable composites with improved mechanical performance. For this reason, I will recommend the present scientific manuscript for further publication in Polymers once all the aforementioned suggestions will be properly fixed.
Answer:
Thanks. The manuscript has been revised according to your comments.

Reviewer 2 Report
The aim of the research was to process poly(lactic acid) composites reinforced with regenerated cellulose fibres (lyocell) into novel filaments and used for 3D printing. A thorough characterisation of the materials was also carried out.
It's a well-written manuscript and the paper will be an interesting study that can contribute to the scope of the Polymers. The manuscript and supplementary materials contains a lot of typographic errors (some highlighted below), which should be corrected.
1. Title, line 11, and 35, supplementary mat.: “polylactic acid”, “Polyethylene glycol”; According IUPAC nomenclature names of polymers whose monomers consist of two words or more are written with parentheses. Should be poly(lactic acid), etc. Please correct.
2. In the abstract, the authors should point out that the composites are not "novel", but the use of them to obtain the filament, as they indicated later. Please correct.
3. Italic format is missing for location prefixes and stereochemical arrangements as “N-” (according IUPAC recommendations). Please correct throughout the text.
4. Line 85: “Poly (lactic acid)”; There is no space between the poly and the parenthesis. Please correct.
5. Line 86: “2.16kg”, also 95, 111, 137, 142, 149, 150, 168, 174, Figure 4, 319, 453, etc.; There should be a space between the variable and its unit. Please correct throughout the text.
6. Line 86: “g.cm-3”; There should be no dot here. A space or half-high dot is used to signify the multiplication of units. Please correct throughout the text.
7. Line 208 and next: Please correct throughout the text typographic errors. According to SI variables should be in italics (e.g. Tg). Please also use subscripts. Please correct throughout the text.
8. Line 244 and next: There should be a space before and after the mathematical operators such as “=”. Please correct throughout the text.
9. Figure 4: “– 10%”; The dash in this case may suggest a minus. Could the authors change the notation?
10. Line 317, also 441, 531: “@break”; Please correct the typographic errors.
11. How the authors explain bimodal melting endotherm?
12. Please correct typographic errors in supplementary materials.
Author Response
Reviewer 2:
Comments and Suggestions for Authors
The aim of the research was to process poly(lactic acid) composites reinforced with regenerated cellulose fibres (lyocell) into novel filaments and used for 3D printing. A thorough characterisation of the materials was also carried out.
It's a well-written manuscript and the paper will be an interesting study that can contribute to the scope of the Polymers. The manuscript and supplementary materials contains a lot of typographic errors (some highlighted below), which should be corrected.
Answer:
Thank you. Your revision will help us improve the quality of our manuscript.
- Title, line 11, and 35, supplementary mat.: “polylactic acid”, “Polyethylene glycol”; According IUPAC nomenclature names of polymers whose monomers consist of two words or more are written with parentheses. Should be poly(lactic acid), etc. Please correct.
Answer:
Thanks. Both documents were corrected accordingly.
- In the abstract, the authors should point out that the composites are not "novel", but the use of them to obtain the filament, as they indicated later. Please correct.
Answer:
Thanks. Corrected.
- Italic format is missing for location prefixes and stereochemical arrangements as “N-” (according IUPAC recommendations). Please correct throughout the text.
Answer:
Thanks. Corrected.
- Line 85: “Poly (lactic acid)”; There is no space between the poly and the parenthesis. Please correct.
Answer:
Thanks. Corrected.
- Line 86: “2.16kg”, also 95, 111, 137, 142, 149, 150, 168, 174, Figure 4, 319, 453, etc.; There should be a space between the variable and its unit. Please correct throughout the text.
Answer:
Thanks. Corrected.
- Line 86: “g.cm-3”; There should be no dot here. A space or half-high dot is used to signify the multiplication of units. Please correct throughout the text.
Answer:
Thanks. Corrected.
- Line 208 and next: Please correct throughout the text typographic errors. According to SI variables should be in italics (e.g. Tg). Please also use subscripts. Please correct throughout the text.
Answer:
Thanks. Corrected.
- Line 244 and next: There should be a space before and after the mathematical operators such as “=”. Please correct throughout the text.
Answer:
Thanks. Corrected
- Figure 4: “– 10%”; The dash in this case may suggest a minus. Could the authors change the notation?
Answer:
Thanks. Corrected
- Line 317, also 441, 531: “@break”; Please correct the typographic errors.
Answer:
Thanks. Corrected
- How the authors explain bimodal melting endotherm?
Answer:
Thanks for the observation. The following sentence was included in the DSC discussion
In addition, the heat treated L10% and L20% composites present a bimodal melting endotherm (Figure 8b) that is attributed to the melting of the metastable α form of PLA (α’) potentially formed during 3D printing [37–41]. The first peak of fusion is related to the simultaneous melting of the primary α crystals formed during printing and recrystallisation of α’ to α crystal form. The second peak of fusion is attributed to the melting of α crystal form generated in the recrystallisation process [42].
- Please correct typographic errors in supplementary materials.
Answer:
Thanks. Corrected

Reviewer 3 Report
The purpose of this study was to examine the properties of polylactic acid composites reinforced with regenerated cellulose fibres when applied to 3D printing. The process of material preparation and characterization align with the conventions in the corresponding field. The results are addition to the existing publication. It can be published after addressing the following issues:
1. Page 3, Section 2.3. It is highly recommended that the author provides a flowchart or images of the production process.
2. Page 4, Section 2.4.3. Same as the previous comment, a flowchart or schematic should be provided for better understanding.
3. Pag3 7, Figure 2. The sub-label of (c) and (d) in the image is lacking.
4. Page 9, Figure 4. Each sub-figure should be labeled and described in the caption. Same for other figures in this paper such as Figure 5 and Figure 6.
5. Page 10. Section 3.1.2. The stress-strain curve of the tests should be provided. I found one in the supplemental material, but, this is one main results of the characterization so it should be included in the main paper. Same for the Page 11, Section 3.2.1.
Author Response
Reviewer 3:
Comments and Suggestions for Authors
The purpose of this study was to examine the properties of polylactic acid composites reinforced with regenerated cellulose fibres when applied to 3D printing. The process of material preparation and characterization align with the conventions in the corresponding field. The results are addition to the existing publication. It can be published after addressing the following issues:
Answer:
Thank you. Your revision will help us improve the quality of our manuscript.
- Page 3, Section 2.3. It is highly recommended that the author provides a flowchart or images of the production process.
- Page 4, Section 2.4.3. Same as the previous comment, a flowchart or schematic should be provided for better understanding.
Answer:
Thank you for the suggestion. We have included a schematic representation for the production of the filaments (Figure 1). For section 2.4.3, we believe the methodology used to extract the fibres are clear and does not require an additional image. In addition, there are already too many Figures in the manuscript that are necessary to have a good understanding of this study.
- Pag3 7, Figure 2. The sub-label of (c) and (d) in the image is lacking.
Answer:
Thank you. Corrected.
- Page 9, Figure 4. Each sub-figure should be labeled and described in the caption. Same for other figures in this paper such as Figure 5 and Figure 6.
Answer:
Thank you. Corrected.
- Page 10. Section 3.1.2. The stress-strain curve of the tests should be provided. I found one in the supplemental material, but, this is one main results of the characterization so it should be included in the main paper. Same for the Page 11, Section 3.2.1.
Thank you for the suggestion. We believe there are already too many Figures in the manuscript that are necessary to have a good understanding of this study. The mechanical properties (including strain at break) are already well represented in Tables 4 and 5, so we think that the stress/strain curves would be better positioned in the supplementary material.
